# Design and Implementation of a Low-Cost Intelligent Unmanned Surface Vehicle

**DOI:** 10.3390/s24103254

**Published:** 2024-05-20

**Authors:** Piyabhum Chaysri, Christos Spatharis, Kostas Vlachos, Konstantinos Blekas

**Affiliations:** Department of Computer Science & Engineering, University of Ioannina, 45110 Ioannina, Greece; p.chaysri@uoi.gr (P.C.); cspatharis@cs.uoi.gr (C.S.); kostaswl@cs.uoi.gr (K.V.)

**Keywords:** USV, autonomous navigation, low-cost USV platform, reinforcement learning

## Abstract

This article describes the design and construction journey of a self-developed unmanned surface vehicle (USV). In order to increase the accessibility and lower the barrier of entry we propose a low-cost (under EUR 1000) approach to the vessel construction with great adaptability and customizability. This design prioritizes minimal power consumption as a key objective. It focuses on elucidating the intricacies of both the design and assembly processes involved in creating an economical USV. Utilizing easily accessible components, the boat outlined in this study has been already participated in the 1st Aegean Ro-boat Race 2023 competition and is tailored for entry into similar robotic competitions. Its primary functionalities encompass autonomous sea navigation coupled with sophisticated collision avoidance capabilities. Finally, we studied reinforcement learning strategies for constructing a robust intelligent controller for the task of USV navigation under disturbances and we show some preliminary simulation results we have obtained.

## 1. Introduction

As the era of artificial intelligence (AI) continues to impact several sectors of the economy and autonomous technologies continue to mature, a new era of marine industry, maritime innovation and sustainability is dawning. The growing field of small surface vehicle technology promises innovative solutions to revolutionize shipping and transport systems, with extensive research focused on unlocking benefits such as improved safety and efficiency. Within this landscape, autonomous systems have emerged as key elements in transportation infrastructure and the creation of intelligent solutions [1]. Researchers are increasingly directing their efforts towards developing control techniques, leveraging AI, in pioneering autonomous marine vehicles. The ability of AI to navigate uncertain and highly constrained dynamical systems is crucial, enabling adaptation to environmental changes and noisy conditions, and facilitating the implementation of optimized decisions.

There are plenty of benefits of making surface vehicles unmanned such as shipping flexibility, reducing costs and minimizing the impact, limitation and cost of human operators [2]. Unmanned surface vehicles (USVs) have been involved in military, research and commercial applications, including surveillance, data collection and sea, surface and space communication hubs [3,4,5,6].

However, the majority of USVs are typically large and expensive, as well as being custom-designed and built for their target application and are not generally available for purchase commercially. Existing commercial USVs are expensive and not fully based on open-source hardware, making it difficult to purchase them and to modify their designs and programming to suit various environments. Moreover, there are no available open-source USV platforms that would allow customization for varied purposes. In general, the above factors make the cost of purchasing commercially available USVs prohibitive for many laboratories and researchers.

In the past few years, the price of the components has decreased and microcontrollers, such as Arduino, are easily obtainable, which makes the construction of such vessels more accessible. There were attempts on low-cost USV [7], such as the small USV for mapping and monitoring coral presented in [8]. The cost without the camera is not so different than our proposed vessel but the operation time is limited to approximately 40 min. The use of low-cost components to enhance a manual USV with autonomous functions has also been explored in [9] to access the precision for performing bathymetric measurements. Another example of a low-cost approach for bathymetric measurements was completed in [10]. The aforementioned works utilized ArduPilot Pixhawk 1 and ArduRover 2.43 respectively for navigation purposes. Although the software is open-source, flexible and easy to use, the integration of more advanced sensors and navigation functions are still limited.

In contrast, the USV platform proposed by our team is completely open-source, from hardware to software. Some parts of the platform were 3D-printed and they can be easily modified and upgraded in terms of both design and programming. Our primary goal was to design and build a low-cost USV, where the parts can be sourced from various online and local vendors and can be replaced easily and at a low cost. This facilitates exploration of new scientific avenues and provides a flexible framework for investigating diverse and interesting research projects. Additionally, to build an easy to carry and deploy USV, we based our design on a small and light floating vessel that is widely available and inexpensive to buy. Because of this design choice, the proposed USV is suitable for applications in confined environments, like lakes or ports, where the wind and wave disturbances are limited. Possible applications could be the water quality monitoring, object localization and identification on a lake surface, seabed inspection in ports and marine life monitoring. The development of an USV for the open sea or the ocean is out of the scope of this paper.

Given our primary focus on machine learning, reinforcement learning (RL), control of autonomous marine platforms, and intelligent agents [11,12,13], we have already delved into the study and implementation of ship navigation algorithms utilizing RL within simulated environments [14,15], and control of autonomous robotic marine platforms [16,17]. Encouragingly, we have achieved promising results, attaining high accuracy and self-learning capabilities. We aim to continue studying and potentially enhancing these algorithms while operating in real-world settings. Additionally, we aspire to develop tailored intelligent agents to address navigation challenges specific to USVs.

Gathering together all the previous findings, we can emphasize the advantages of the proposed USV platform as follows:It is designed with cost-effectiveness in mind, offering a lower initial purchase cost compared to many existing models in the market. Through innovative design features and efficient energy management systems, the proposed USV minimizes operational expenses, resulting in long-term cost savings.It has the flexibility to incorporate cutting-edge technologies such as an autonomous navigation system and remote sensing capabilities, offering enhanced situational awareness and mission effectiveness, expanding its applicability to covering multiple needs.Although the system has some limitations mostly due to its size, it demonstrates adaptability and reliability across a spectrum of operational scenarios, and it covers the capability to perform significant applications, such as environmental monitoring, water quality and maritime surveillance.The proposed USV has lightweight construction materials allowing extended endurance, prolonged missions without the need for frequent refueling or recharging, as well as the easy integration and customization of various sensor systems.

The structure of our manuscript is as follows. Section 2, Section 3 and Section 4 present a meticulous analytical description of the fundamental buildings blocks constituting the proposed USV platform. We delve into comprehensive presentations of each component, clarifying their roles as well as their interactions with the system. Following this, we give details about the procedures required for the assembly and rigorous testing of the platform, ensuring precision and reliability in its construction. Then, Section 5 illustrates the reinforcement earning agent we have created for automatically controlling and navigating the proposed USV system. To showcase the platform’s basic functionalities, Section 6 demonstrates its main capabilities in various simulated and real-world scenarios. In addition, we illustrate the efficiency of the decision support system integrated into the platform, highlighting its role in enhancing the system’s autonomy behavior and adaptability in dynamic marine environments. Finally, Section 7 concludes the manuscript with findings and future directions for further study.

## 2. Hardware Architecture

This section outlines the hardware and components that were used in the construction of the vessel, starting with a brief outline of the components, and then discusses each of them in detail in the following subsections.

We utilized a small kayak as a base for the floating platform to simplify the building process. Subsequently, we integrated an electric motor for propulsion and mounted a servo for directional control atop it. The propulsion was an electric trolling motor for kayak to skip the waterproofing process for the thruster and simplify the building process even further. Furthermore, the control unit was a microcontroller, specifically an Arduino Mega, chosen for its versatility in communicating with both sensors and actuators. The electronic speed controller for the motor and the servo receive commands from the microcontroller.

### 2.1. Hull

The hull, or the main body of the vessel, has been selected within the rules of the Ro-boat Race competition which stated that the boats must be under 2.5 m in length. Our primary consideration was ensuring that the vessel was sufficiently robust to withstand the sea and waves while also adhering to the competition’s regulations. After careful consideration, we opted for the Lifetime Wave, a small kayak designed for children, as it met our requirements (see Figure 1). Additionally, this kayak offers ample space to accommodate all the necessary equipment. Table 1 presents the specifications of the selected kayak.

### 2.2. Propulsion

The propulsion is provided by an electric trolling motor and is normally powered by a 12 V deep cycle battery. The design of such motor is a thruster motor with propeller on a pole that can be attached to the back of a kayak or a small boat. We chose the Haswing W20 trolling motor manufactured by Ya Tai Electric Appliance Co., Ltd. Guangdong, China. for this work shown in Figure 2a, due to its low price and availability.

The steering is controlled by a high-voltage, hi-speed servo, AGFRC A280BVSW, which is shown in Figure 2b. This servo was provided to us by AGF-RC Electronic Technology Co., Ltd., Guangdong, China. at no cost. It was originally designed for 1/5 scale R/C cars and it was chosen for being waterproof (IP67 certified) and having high torque with immediate response.

The steering range was limited to $120^{\circ }$ overall or $60^{\circ }$ to each side. The thrust and steering range are depicted in Figure 3. The specifications of the motor and servo can be found in Table 2. By using a single thruster and a servo to propel and steer the vessel we can greatly reduce the power consumption.

### 2.3. Sensors, Computing Unit and Power

The vessel obtains its position and orientation data from GPS, gyroscope, accelerometer and magnetometer (compass). Arduino was utilized to receive and process the sensor data, which includes GPS position, angular orientation, angular velocity and linear acceleration. In addition to the positioning data, the vessel can also perceive its surroundings using a forward-facing camera, which is connected to the Raspberry Pi for image processing. A radio control transmitter was used to remotely operate the vessel as well as turning the autonomous functions on and off. The overall connection between sensors, computing unit and propulsion is depicted in Figure 4. The sensor unit consisted of a GPS, an integrated IMU sensor and a Micro SD card writer for recording the sensor data. It must be noted that the thruster was an electric DC motor that can be controlled by an electronic speed controller (ESC) from radio control vehicles. This can subsequently be controlled via PWM signal. The steering servo was also a part of radio control vehicle. Once again, this can also be controlled via PWM signal. This was to simplify the autonomous agent to control the vessel as both thruster and steering can be controlled through Arduino as well as the GPIO library on the Raspberry Pi. We chose to utilize one thruster and steering along with a low-powered computing unit in consideration of power consumption. While sacrificing speed and maneuverability, having only one engine reduced the power consumption significantly.

To power the propulsion and computing unit, 4-cell lithium polymer (4S LiPO) batteries were used. Each battery had the nominal voltage of 14.8 V and all batteries were connected in parallel, which allowed us to have different battery configurations between 2 and 4 batteries depending on the required operation time. We chose to increase the battery in pair because of the battery placement in order to maintain the weight distribution. LiPO was chosen due to its low cost, low weight and high energy density (6000 mAh 4S LiPO batteries). The power was provided to the computing unit through a DC-to-DC converter, a step down to 5 V 5 A.

## 3. Building Process

In this section we discuss the process of converting a kayak into an autonomous vehicle. To achieve this, it was necessary to securely mount the propulsion system on the hull. Additionally, the steering system required conversion from manual lever operation to a servo steering mechanism. Finally, all electronics and batteries needed to be securely attached to the hull with waterproofing measures in place.

### 3.1. Motor Mounting Solution

The trolling motor, or thruster motor, needed to be mounted on a flat surface. However, the area at the back of our kayak was curved, making it impossible to attach the bracket for the thruster motor. The solution wass to use a 50 × 50 cm Plexiglass sheet with a thickness of 5 mm, securely screwed onto the hull to provide a flat surface for mounting the motor bracket. We then modified the trolling motor mount by sawing off a portion to create mounting points, enabling us to attach it to the Plexiglass sheet. The design of the mounting points and the resulted motor mounting solution can be seen in Figure 5.

### 3.2. Steering Mechanism

The reason we utilized the original motor mount was to keep its pivot point intact in order to use it as a part of steering mechanism. The original steering mechanism was a shaft with locking collar to stop the motor from dropping down. This shaft can be rotated for steering using the steering handle located on the top, as shown in Figure 2a.

We converted this action from the driver physically moving the handle to using a servo to steer. We started by making the servo mounting bracket to make it sit upright. This is needed in order to hold the steering mechanism down to the plate with absolute rigidity. The picture of the servo bracket is shown in Figure 6.

We applied the steering mechanism using two pull rods (of size M5), which were connected directly from the servo to the motor shaft. The design of the servo arm extension and the steering bracket are depicted in the Figure 7. We found that the Plexiglass or acrylic sheets provided a great solution whenever a flat surface was needed. Furthermore, once the heat was applied, the sheet could be bent to the desired angle. We used this technique for the steering part, as depicted in Figure 8a. It was then placed on the trolling motor shaft. This was later substituted with 3D printing for more durability and ease of construction. We limited the steering range to $120^{\circ }$ to prevent the steering mechanism from locking up at the maximum steering angle. With the motor and steering system mounted on the kayak, it was been converted to a remotely operated vehicle. The dimensions of the vessel with its engine are depicted in Figure 8b.

### 3.3. Electronics Placement

The electronics and batteries were placed in water-proof boxes mounted along the craft with the consideration of the weight distribution. Their placement is depicted in the Figure 9. The boxes at the front contain batteries (see Figure 9a), while the box shown in Figure 9b contains speed control, microcontroller and sensors. As the batteries constituted the second heaviest component in the vessel, following the motor and steering mechanism, we positioned them at the front to offset the weight of the heavy rear section. Moreover, we drilled several holes on the boxes to thread the power wires in, and then we used hot glue to form the seal around the holes where the wires come out. The fully assembled vessel, along with our team members, is depicted in Figure 10.

### 3.4. Budget

Cost was a significant concern for us, as this project did not receive support from any grants. Instead, funding for the project was provided by our own team members. Additionally, time constraints were a key consideration, necessitating the timely arrival of every component. To optimize our budget, we meticulously selected each component based on its price-to-performance ratio, taking into account factors such as its origin and estimated delivery time.

Our procurement process began by sourcing all the necessary parts from a diverse range of local and overseas vendors. In addition to the hull and propulsion system, mechanical components like ball joints were also essential to complete the build. While the kayak, trolling motor and the electronic speed controller (ESC) could be purchased within Greece, we could not find the battery with high capacity at an affordable price. Furthermore, the ball joints were also not available within Greece, hence these parts were purchased from overseas vendors to keep the spending low.

The waiting time for the parts to arrive varied depending on the vendor. Generally, products from local vendors typically arrived within 3 days, whereas orders from overseas vendors ranged between 12 to 26 days for delivery. The total cost, inclusive of shipping fees, is detailed in Table 3. It is important to note that the cost of the servo is not included in this calculation as it was generously provided to us at no cost. The current market price of the servo stands at EUR 315.86. However, for those aiming to maintain a lower budget, a substitute servo can be acquired for approximately EUR 64.

## 4. Software Architecture

The software functionalities of the proposed platform are as follows:Autonomous waypoint following using GPS and compass.Visual-based obstacle avoidance using camera.Remotely operated (teleop) via R/C transmitter.Remotely switch between manual control, waypoint following, waypoint following with obstacle avoidance and a function to skip a waypoint in case it is out of reach.

The remote control was a four-channel R/C transmitter. For the teleop function, channel 1 and 2 were used for steering and throttle respectively. channel 3 was used to activate the autonomous waypoint-following algorithm. The next waypoint could be skipped by alternating channel 3 between high and low. Finally, channel 4 was used to activate the visual-based obstacle avoidance. Next, we describe the navigation algorithm for the following waypoints.

### Navigation Algorithm

The autonomous waypoint following algorithm is depicted in Figure 11, where

ψ: the orientation of the vessel.$\theta_{desired}$: the desired orientation.$\theta_{error}$: the difference between the vessel orientation and the desired orientation.

The vessel’s latitude and longitude $V_{lat},V_{lon}$ were acquired via GPS, while the orientation ψ was obtained via compass. The waypoints’ latitude and longitude $WP_{lat},WP_{lon}$ were manually entered into the program; therefore, the Arduino needed to be flashed each time a waypoint position was changed. The vessel navigated from any position to the next waypoint in a straight line and the autonomous function cycled through all waypoints and then repeated the process until it was deactivated. Each waypoint was considered completed if the distance between the vessel and the waypoint was under a threshold of 2.2 m. Algorithm 1 describes the framework for waypoint following.
**Algorithm 1:** Waypoint following (update rate: 2 Hz)
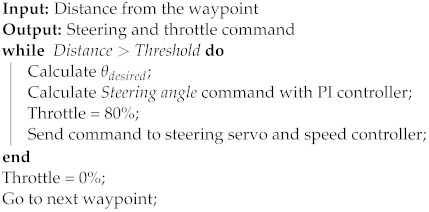


It must be noted that in the above algorithmic structure, *Threshold* denotes the target distance from the waypoint coordinates. On the other hand, the distance function to follow the waypoints is calculated using the Haversine Formula [18]:(1)$$\begin{array}{cc} \Delta_{lat} & =WP_{lat}-V_{lat}\\ \Delta_{lon} & =WP_{lon}-V_{lon}\\ a & =\sin^{2}\left(\frac{\Delta_{lat}}{2}\right)+\cos\left(V_{lat}\right)\cdot \cos\left(WP_{lat}\right)\cdot \sin^{2}\left(\frac{\Delta_{lon}}{2}\right)\\ c & =2\cdot \mathrm{atan}2\left(\sqrt{a},\sqrt{1-a}\right)\\ R & =Radius\;of\;the\;Earth\\ Distance & =R\cdot c \end{array}$$

Moreover, the desired orientation is obtained as follows:(2)$$\begin{array}{cc} y & =\sin\left(\Delta_{lon}\right)\cdot \cos\left(WP_{lat}\right)\\ x & =\cos\left(V_{lat}\right)\cdot \sin\left(WP_{lat}\right)-\sin\left(V_{lat}\right)\cdot \cos\left(WP_{lat}\right)\cdot \cos\left(\Delta_{lon}\right)\\ \theta_{desired} & =\mathrm{atan}2(x,y) \end{array}$$

Finally, the *Steering angle* is controlled by a PI controller:(3)$$\begin{array}{cc} \theta_{error} & =\theta_{desired}-\psi \\ \mathrm{Steering}\mathrm{angle} & =K_{p}\cdot \theta_{error}+K_{i}\int_{0}^{t}\theta {\left(t\right)}_{error}\;dt \end{array}$$

The collision avoidance utilized the SSD MobileNet V3 with COCO dataset (https://github.com/zafarRehan/object_detection_COCO (accessed on 28 March 2024)) for object detection using a camera running on Raspberry Pi4 Model B.(https://www.raspberrypi.com/products/raspberry-pi-4-model-b/ (accessed on 28 March 2024)) We chose the MobileNet for this work because we found that it was able run on Raspberry Pi at an adequate speed with good detection. It must be mentioned that we did not train our own model, but instead we utilized a pre-trained one to speed up the building process. In order to mitigate the false detection of the object, we experimented with different classes that the buoys would be recognized as. We concluded that the collision detection will only recognize three classes or categories, as follows: boat, person and airplane. The detection was executed every 2 s so that in case an obstacle was not recognized the first time, it would be in the next step. The objects detected using this method are depicted in Figure 12. The camera resolution was set to 640 × 480 and the boat was cropped out before the collision avoidance calculation process. At that stage, it was only used for the horizontal position and perceived size (occupying pixel count) of the object. Algorithm 2 presents the structure of the collision avoidance steering command framework. It has to be noted that the algorithm only considers the largest detected object in view.

The Raspberry Pi was responsible for sending the aforementioned steering command to the Arduino, which was then combined with the waypoint following algorithm, resulting in the final steering command. The diagram of the autopilot system is depicted in the Figure 13. The codes used in this work are available in the provided link (https://github.com/PiyabhumC/Ro-Boat_2023 (accessed on 28 March 2024)).
**Algorithm 2:** Collision avoidance (camera running at 2 FPS)**Input**: Largest detected object’s position $O_{x}$ and width $O_{w}$**Output**: Steering command [−60,60]Direction=1,if Ox<320/*(rightturn(*/−1,Otherwise/*(leftturn(*/*Steering angle* = max(O_*w*_, 60);return Steeringangle×Direction

## 5. Reinforcement Learning Agent for USV Navigation and Control

The recent literature has introduced various methods for marine platform navigation utilizing reinforcement learning (RL) techniques [14,19]. In the realm of USVs, deep RL (DRL) methodologies have gained a lot of attention. For example, Ref. [20] used the deep deterministic policy gradients (DDPG) [21] algorithm to determine the desired heading with an adaptive sliding mechanism regulating both the heading and velocity to ensure effective path-following for the USV. Additionally, Ref. [22] introduced a DRL algorithm tailored for collision avoidance among multiple USVs operating in complex scenarios, Ref. [23] proposed a novel collision avoidance method using DRL, and [24] developed evasion strategies for USVs in multi-obstacle environments.

Generally, in RL [13], the formulation of problems often relies on the framework of Markov Decision Processes (MDPs) (Figure 14), which offers a way to model sequential decision-making problems. For instance, the navigation and control scenario examined in this study can be effectively characterized as an MDP. Specifically, the agent interacts with the environment by collecting observations (states) and performing appropriate actions for the current state. Subsequently, upon executing these actions, the environment transitions into a new state, $s_{t+1}$, and receives a reward, $r_{t}$. This process iterates until a terminal state is reached, and the standard RL objective is to maximize the sum of expected rewards:(4)$$\sum_{t}E_{\left(s_{t},a_{t}\right)∼\rho_{\pi }}\left[r\left(s_{t},a_{t}\right)\right]\;,$$
where $\rho_{\pi }$ denotes the joint distribution over states and actions based on policy $\pi (a_{t}|s_{t})$.

Within the realm of RL, value-based methods [25] represent a framework that emphasizes on the estimation of action values over different states. Optimal policy $\pi^{*}$ is derived by iteratively improving the value function according to the Bellman equations. Learning algorithms such as Deep Q-Network (DQN) [26] and its variations operate within this paradigm, trying to converge towards an optimal value function and using it to construct an optimal policy. Nevertheless, these approaches struggle in adapting to scenarios featuring high-dimensional or continuous action spaces. In contrast, policy-based methods focus on directly optimizing the policy function π(a|s) to maximize the expected rewards. Learning schemes like Proximal Policy Optimization (PPO) [27], Trust Region Policy Optimization (TRPO) [28] and their variants fall under this category, leveraging gradient ascent to enhance policy parameters. Unlike value-based methods, which concentrate on policy evaluation and improvement separately, policy-based approaches continuously refine the policy through iterative updates.

However, both approaches face some limitations; hence, actor-critic methods [29] have emerged as hybrid solutions in RL. These approaches combine the strengths of both paradigms to facilitate more robust learning. In such a framework, the actor network selects actions under a stochastic manner, while the critic network provides value estimates based on the actions chosen by the actor. Subsequently, the actor network updates its parameters based on the feedback from the critic network, thereby enhancing the policy over time. A state-of-the-art advancement in this domain is the Soft Actor–Critic (SAC) [30,31] algorithm, which further refines the actor–critic architecture to address issues related to sample efficiency and exploration in continuous action spaces. Leveraging an off-policy update mechanism and an experience replay, SAC addresses the challenge of sample inefficiency commonly encountered in RL tasks. Moreover, SAC introduces entropy maximization as an additional objective, promoting exploration that aids in discovering diverse strategies and contributes to the stability of the learning processes in complex environments. Formally, SAC optimizes the following objective:(5)$$\sum_{t}E_{\left(s_{t},a_{t}\right)∼\rho_{\pi }}\left[r\left(s_{t},a_{t}\right)+\alpha H\left(\pi \left(\cdot |s_{t}\right)\right)\right]\;,$$
where α is a temperature weight of the entropy H of the policy that is given by
(6)$$H\left(\pi \left(\cdot |s_{t}\right)\right)=-E_{\pi }\left[log\pi \left(\cdot |s_{t}\right)\right]\;.$$

The primary goal of SAC is to discover the optimal policy, represented as $\pi^{*}$, which aims to maximize the entropy-augmented objective function. To achieve this, it requires the calculation of soft Q and value functions, which satisfy the following soft Bellman equations:(7)$$Q\left(s_{t},a_{t}\right)=r\left(s_{t},a_{t}\right)+\gamma E_{\left(s_{t+1}\right)∼p\left(s_{t+1}|s_{t},a_{t}\right)}\left[V\left(s_{t+1}\right)\right]\;,$$
where value function is obtained by
(8)$$V\left(s_{t}\right)=E_{a_{t}∼\pi }\left[Q\left(s_{t},a_{t}\right)-\alpha \;log\pi \left(a_{t}|s_{t}\right)\right]\;.$$

We can evaluate the soft *Q* value of a fixed policy π by applying the Bellman equation at each time step, which is called soft policy evaluation in the soft policy iteration. This is achieved by minimizing the expected Kullback–Leibler (KL) divergence between the policy and the soft *Q*-function. However, applying this obective directly to large continuous state–action spaces is infeasible.

Rather than executing the policy iteration until convergence, SAC employs parameterized neural networks as function approximators for both the *Q*-function and the policy. Specifically, SAC comprises three networks: (a) actor network, (b) critic network and (c) value function network. The critic (or soft *Q*-function) network $Q_{\phi }\left(s,a\right)$, parameterized by ϕ, estimates the action-value function, while the value network $V_{\psi }\left(s\right)$, parameterized by ψ, calculates the state-value function. The soft *Q*-function parameters ϕ can be optimized by minimizing the squared soft Bellman residual given by
(9)$$J_{Q}\left(\phi \right)=E_{(s_{t},a_{t})∼D}\left[\frac{1}{2}{\left(Q_{\phi }\left(s_{t},a_{t}\right)-\left(r\left(s_{t},a_{t}\right)+\gamma E_{s_{t+1}}V_{\psi }\left(s_{t+1}\right)\right)\right)}^{2}\right]\;.$$

Furthermore, SAC employs two target networks similar to the previous ones to ensure algorithm convergence.

The actor (or policy) network $\pi_{\theta }\left(a|s\right)$, with weights θ, selects actions in a stochastic manner using the reparameterization trick and updates its weights by minimizing the expected KL divergence:(10)$$J_{\pi }\left(\theta \right)=E_{s_{t}∼D}\left[E_{a_{t}∼\pi_{\theta }}\left[\alpha log\pi_{\theta }\left(a_{t}|s_{t}\right)-Q_{\phi }\left(s_{t},a_{t}\right)\right]\right]\;.$$

First, in our study of integrating an RL agent into a navigation and control task, we defined the state space, which encompasses several crucial factors, including the linear velocity of the USV and its orientation. Additionally, the distance between the USV and the goal, along with the angle of the goal relative to the USV’s orientation, provide essential spatial context for the navigation task. Moreover, we incorporate two proximity sensors indicating the distance to the closest obstacle and the angle of the obstacle in relation to the USV’s orientation, which are crucial for collision avoidance. Furthermore, environmental factors like wind velocity and its direction relative to the USV’s orientation are integrated into the state space to account for external noise sources on navigation.

On the other hand, the action set represents the choices available to the USV to navigate its environment. These actions include specifying the desired velocity (in m/s), enabling the agent to regulate its speed dynamically based on the circumstances and the desired steering angle of the USV within a range of ±45 degrees. It is worth noting that the actions are assumed to be processed by a low-level controller, ensuring precise execution of the commands issued by the reinforcement learning agent. This integration of high-level decision making with low-level control mechanisms is crucial for achieving robust and efficient autonomous navigation in complex environments.

The reward function evaluates the chosen action’s effectiveness based on the current USV state. We have designed an efficient reward function that prioritizes two objectives: eliminating collisions between our USV and static obstacles, and minimizing travel time. Calculating the reward function first checks if the USV reaches a terminal state, meaning that it either arrives at its destination or collides with a static obstacle. In these cases, the agent receives a constant reward: positive for successfully reaching the goal, or negative as a penalty for collision.

In any other case, the reward function encourages the USV to reach its goal quickly, while avoiding collisions with static obstacles. If there is no obstacle ahead, the agent receives a reward illustrated by the following equation:(11)$$r_{t}=-1+velocity-distance\left(s_{t+1}⟼goal\right)\;,$$
that is a combination of three factors: a small penalty (−1) to discourage multiple steps, a bonus for higher velocity and a relative penalty based on the distance to the goal. If there is a static obstacle detected, the base reward remains the same, however, a punishment factor is added to the overall reward, when a static obstacle is very close. This scheme discourages the agent from getting too close. For obstacles farther away, the punishment is proportional to the obstacle’s distance and the agent’s velocity. The closer the obstacle and the faster the agent, the higher the penalty. This encourages cautious navigation around obstacles.

## 6. Experimental Results

The proposed USV and developed algorithms are extensively tested in both simulated and real-world environments. The results are presented and discussed in this section. We aim to showcase the efficacy and adaptability of our approach across various scenarios, comparing its performance in both real-world settings and simulated conditions.

### 6.1. Real-World Experiments

The primary experimental tests of the proposed platform were conducted in Lake Pamvotis, situated in Ioannina, in north-west Greece. More specifically, the results of our initial tests has proven that the vessel functions properly under calm weather condition. The draft measured from the tests is 8 cm and the top speed recorded from the GPS is 6.4 km/h. It is to be emphasized that the results were obtained on a fresh water lake and under calm weather condition with low wind. A photo of the test environment, in Lake Pamvotis, is shown in Figure 15a. Through the test, we discovered that the battery and electronic compartments were prone to overheating when exposed to direct sunlight. This issue was addressed by implementing heat shields to protect the heat-sensitive areas. Furthermore, the waypoint skipping function proved to be a valuable safety feature which can be served to divert the vessel somewhere else or returning it to the origin point. The main conclusions from these initial trials are the following:The USV was able to follow the waypoints successfully.Due to the pondweed presence, some waypoints had to be skipped because they were unreachable. But all the systems worked as intended.The achieved top speed is equal to 1.64 m/s (3.19 knots) in autonomous mode and 1.78 m/s (3.46 knots) with teleop.The operation time with four batteries was approximately 160 min with enough left to return to base safely.

The GPS log from the test is depicted in the Figure 15b.

Additionally, we participated in the 1st Ro-boat Race (Autonomous Robotic Vessels Competition) aiming to rigorously test the capabilities of our USV under various challenging conditions to assess its performance and reliability. The race took place on 12 July 2023 in Syros, Greece (https://smartmove.aegean.gr/roboat-race/ (accessed on 28 March 2024)), see Figure 16. The competition was divided into three races. The first one was the speed race. Each USV was required to traverse from the starting position then go around other two buoys and back to the starting position to complete a timed lap. The competitors went on the course one at a time. The GPS coordinates of all the buoys were given at the time of the race. We managed to achieve second place in this category. The collected data, during the race, provide us with an insight of the actions and performance of the vessel (see Figure 17). We managed to achieve a top speed of 1.67 m/s or 3.24 knots and the average speed was 1.47 m/s or 2.86 knots with little to no deviation from the shortest path available, which in this case was a straight line.

The second race was the collision avoidance race, where all the competitors were on the course at the same time. The race was between two lines, starting and u-turn points, while the four GPS coordinates were given beforehand. The obstacle buoys were floating randomly between the lines without any prior positioning information. Each competitor had to navigate through the buoys as well as other competitors within the course. The race was one lap from the starting line to the u-turn line and back to the starting line again. The winner was the one that completed the lap the fastest with minimum contact with the obstacles. We took the first place in this category by being the fastest and without making any contact with the obstacles. The performance of this race can be seen in the log depicted in Figure 18. The steering command provided by the collision detection seemed to be erratic in the collected data. Even though the vessel managed to avoid all obstacles, this behavior was not ideal and could be improved further.

The final race was the endurance race, where all the competitors were traversing between two buoys. The one that completed the most laps within the time limit would be the winner. In this category, we secured the second position by completing 19 laps within 26 min. In the data provided by log (Figure 19) the vessel managed to traverse between two waypoints repeatedly and at a constant pace. The average velocity was 1.13 m/s or 2.2 knots. This was slower than the speed race because we decreased the maximum throttle output in order to conserve the battery. All the GPS logs from the races are depicted in Figure 20.

More details about the race results and the competition are published on the official website (https://smartmove.aegean.gr/events/1st-aegean-roboat-race/ (accessed on 28 March 2024)). Our final ranking was second place. Additional materials from our participation in this competition, such as videos and photos, can be found in the provided link (https://drive.google.com/drive/folders/18uuBA1z184QwKC_u4UmGBHdoCrk3HvhV?usp=sharing (accessed on 28 March 2024)).

### 6.2. Simulation Results

We evaluated the performance of the proposed reinforcement learning framework by conducting simulations in an environment closely resembling the one utilized in our previous research [15]. The simulation incorporates both the kinematic and dynamic model of the USV, along with simulated wind and wind-generated wave disturbances. In this study, we constrained the capabilities of the simulated vessel to closely match those of its real-world counterpart in terms of linear velocity. Subsequently, we replicated the geographic area of the island of Syros, Greece within the simulator, shown in Figure 21. The starting point is in the Ermoupoli’s (capital of Syros island) swimming basin and the goal is at the main port area. The prevailing wind direction is Northeast or 225 degrees.

Experiments were made following two scenarios concerning alternative levels of wind velocity: low-wind condition with wind velocity equal to 4 knots and high-wind condition with 7 knots. The purpose of these experiments was to train the USV to navigate out of the designated swimming area and traverse the open sea to reach a specified goal on the opposite side, all while contending with various environmental disturbances.

The learning curves of the reward in terms of the mean reward during navigation paths as obtained by the agent are presented in Figure 22; Figure 23 illustrates some typical navigation paths obtained following the learned policy of the RL agent in both experimental scenarios. One can observe that the generated trajectories are optimal in terms of shortened distance, proving not only the ability and efficiency of the trained agent’s policy, but also its fidelity and stability.

Additionally, Table 4 presents preliminary findings from the agent operating within our simulated environment, detailing statistics gathered from 1000 executions of the agent’s policy in both scenarios. The results are promising, indicating the effectiveness of our approach in enabling autonomous and secure navigation for the USV across a range of environmental conditions. The observed behavior of the agent showcases its ability for thorough exploration, adaptation to real-time fluctuations in environmental dynamics and avoidance of static obstacles.

Lastly, it is important to note that in this study, the reinforcement learning (RL) agent has not been deployed onto the physical hardware. This decision stemmed from the absence of a computing system capable of managing the computational demands necessary for the agent’s operation during the competition, which was the primary purpose for constructing this Unmanned Surface Vehicle (USV).

## 7. Conclusions

This manuscript presents an integrated USV system, meticulously developed within our laboratory at the Department of Computer Science and Engineering, University of Ioannina, Greece. Representing an economical solution, this platform emerges as an ideal instrument for validating concepts and algorithms in marine robotics research. Its innovative design meets the demand for cost-effective marine robotics platforms, finely tuned to the requirements of research endeavors in robotics. Notably, the system’s exceptional flexibility empowers researchers with the ability to swiftly and effortlessly modify, enhance and reprogram its functionalities as required. There is plenty of space left on the hull for future expansion such as lidar and radar as well as vacant pins on the Arduino for more sensors. The USV can also take more payload which can be utilized for more batteries and better computing system for RL agent integration. This adaptability ensures that researchers can seamlessly tailor the platform to suit evolving research needs, allowing for efficient experimentation and innovation in the field of marine robotics. Furthermore, a RL agent has been developed for USV navigation and control under disturbances, which can be integrated into the presented USV. Its simulated performance proves to be reliable with excellent static obstacle avoidance capability. This could be utilized as a marine navigation algorithm for better efficiency and safety.

Finally, many concepts and methodologies employed in the proposed results, such as the reinforcement learning framework and the simulation environment, can be adapted for Networked Marine Surface Vehicles (NMSVs) [32,33], since the the underlying principles of autonomous navigation and adaptation to environmental disturbances in the case of individual USVs remain relevant. For this purpose, we can extend the proposed framework in building collaborative multi-agent systems with sharing data and coordinating actions to achieve common objectives [34,35]. Also, we can possibly consider issues about scalability and adaptability for operating in dynamic marine environments, exploring the capability of NMSVs to dynamically adjust their behavior and strategies based on real-time information exchange within the network. The above framework constitutes a possible direction for further research and a way of providing promising results.

## Figures and Tables

**Figure 1 sensors-24-03254-f001:**
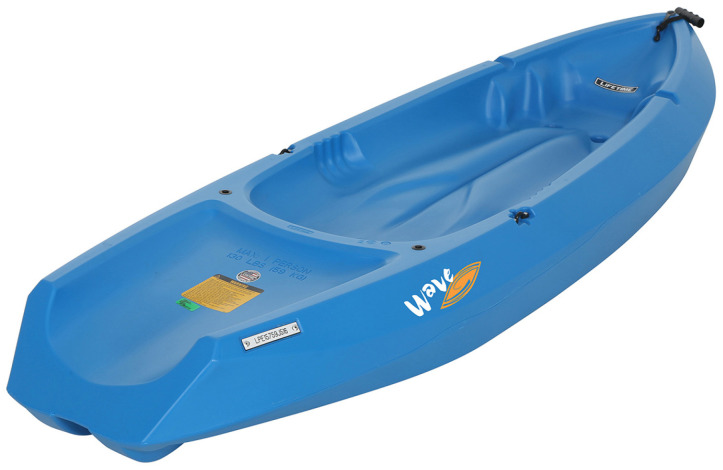
The vehicle chassis we utilized was a simple yet durable Lifetime Wave kayak, chosen for its affordability and adaptability to our demands.

**Figure 2 sensors-24-03254-f002:**
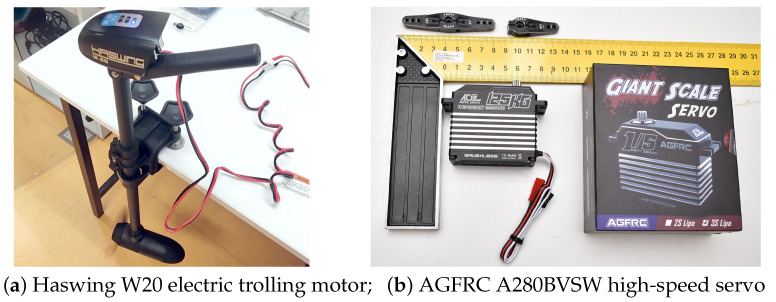
Electric parts of the platform concerning propulsion and steering.

**Figure 3 sensors-24-03254-f003:**
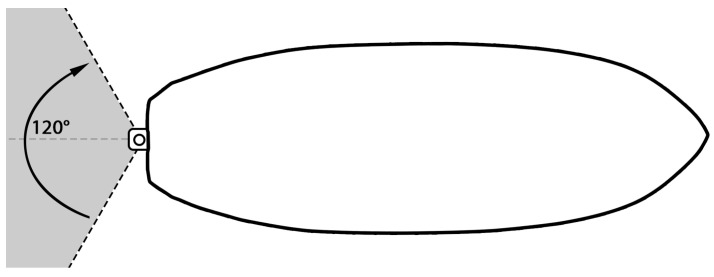
Steering control range of the vessel.

**Figure 4 sensors-24-03254-f004:**
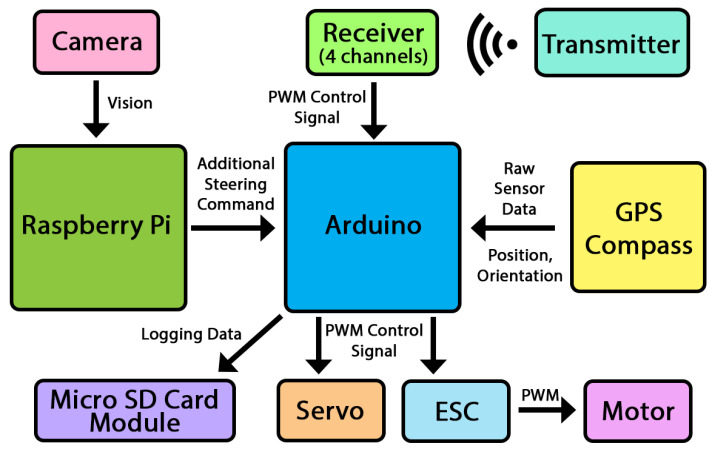
The connection schematic between sensors (GPS, accelerometer, gyroscope, magnetometer), Arduino, Raspberry Pi, electronic speed controller and servo.

**Figure 5 sensors-24-03254-f005:**
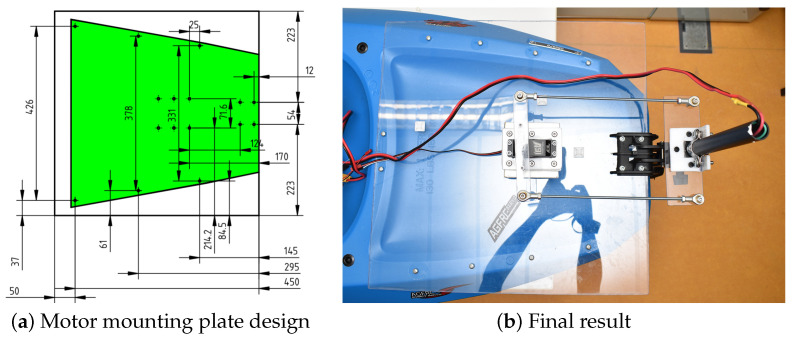
The designs of the motor mounting plate with screw placements template for 50 × 50 cm plexiglass sheet (**a**) and the final result from the design (**b**).

**Figure 6 sensors-24-03254-f006:**
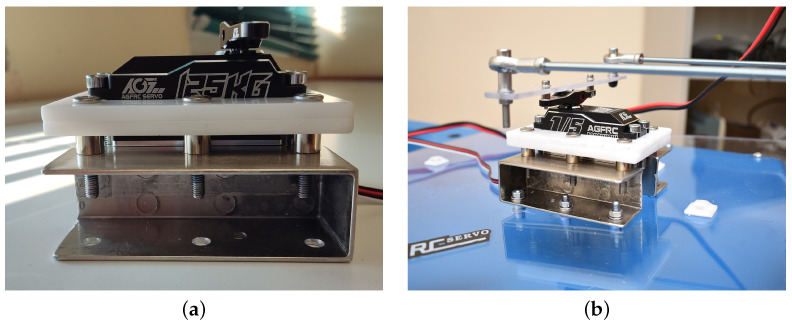
Servo mounting bracket (**a**) for mounting the steering system to the flat surface at the back of the vessel (**b**).

**Figure 7 sensors-24-03254-f007:**
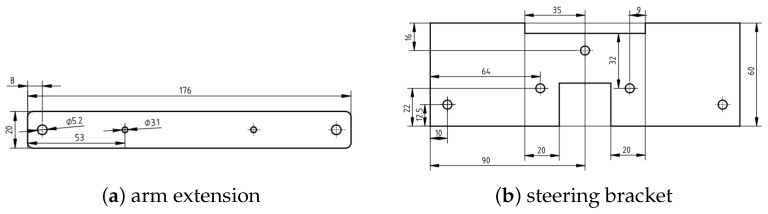
The designs of the servo arm extension (**a**) and steering bracket on the motor shaft (**b**).

**Figure 8 sensors-24-03254-f008:**
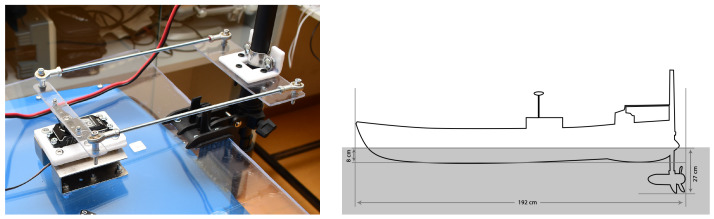
The final version of the steering assembly using ball joints and the dimensions of the vessel.

**Figure 9 sensors-24-03254-f009:**
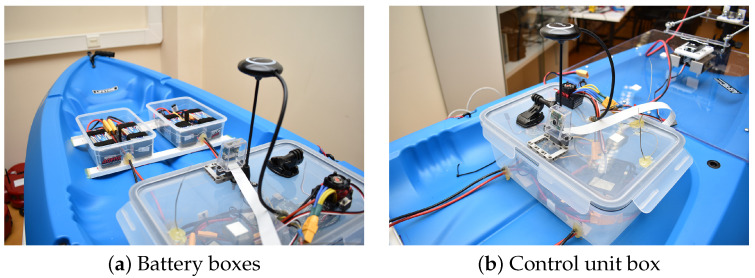
Placing the electronics parts of the vessel.

**Figure 10 sensors-24-03254-f010:**
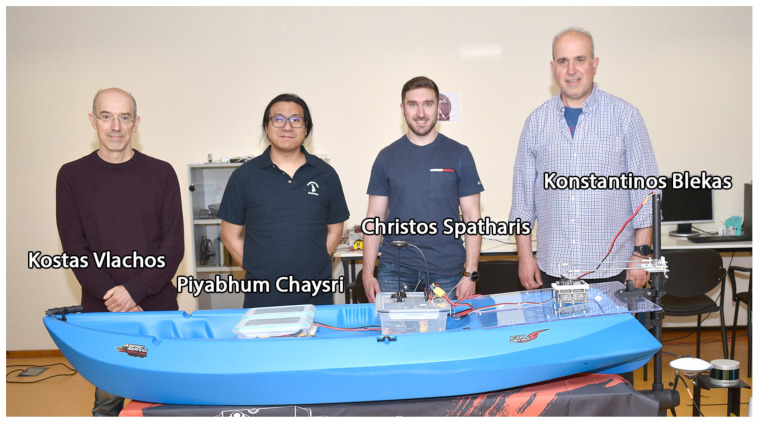
The fully assembled USV with our team members.

**Figure 11 sensors-24-03254-f011:**
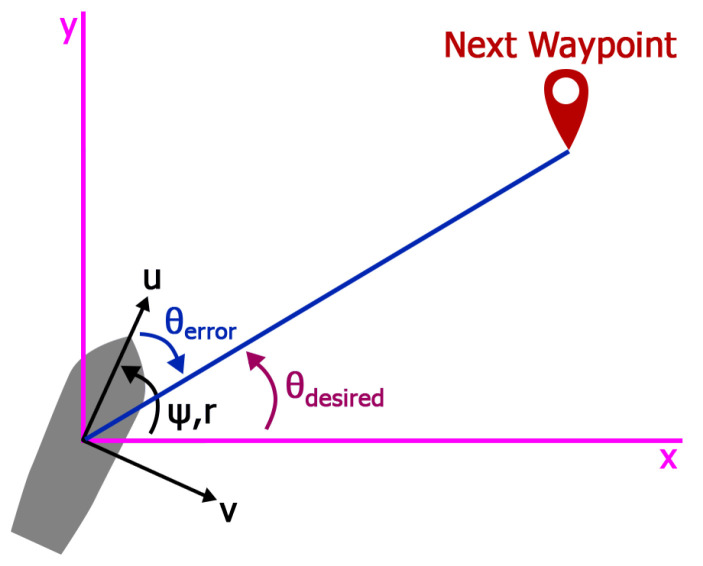
Navigation algorithm used in this vessel.

**Figure 12 sensors-24-03254-f012:**
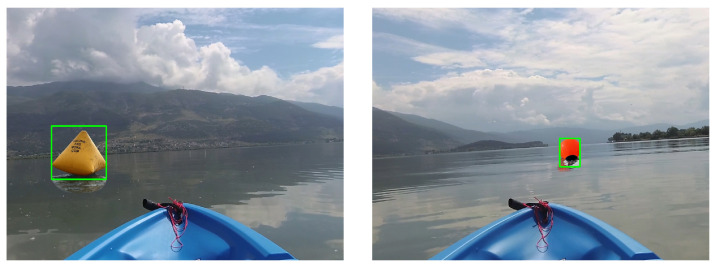
The view from the forward-facing camera with object detection.

**Figure 13 sensors-24-03254-f013:**
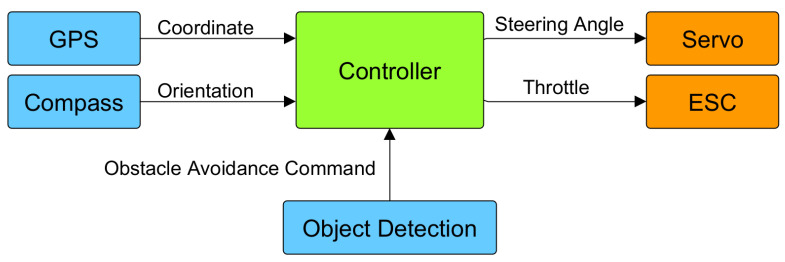
Autopilot system utilizes GPS, compass and extra steering signal from the object detection.

**Figure 14 sensors-24-03254-f014:**
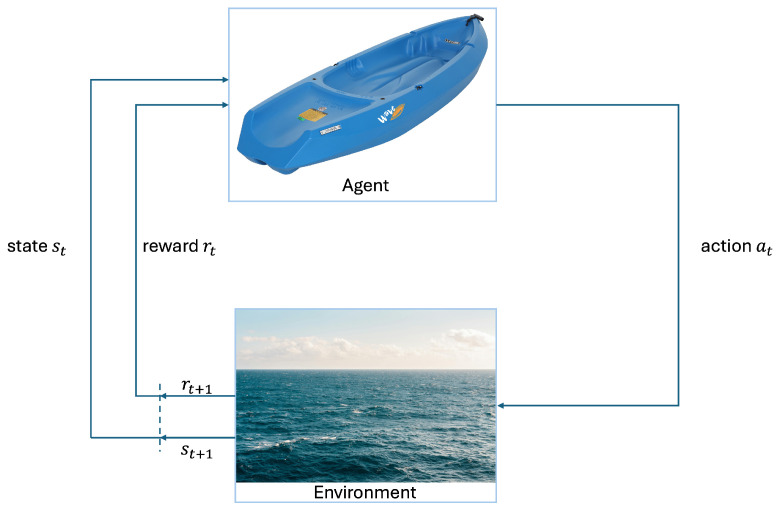
Markov decision process.

**Figure 15 sensors-24-03254-f015:**
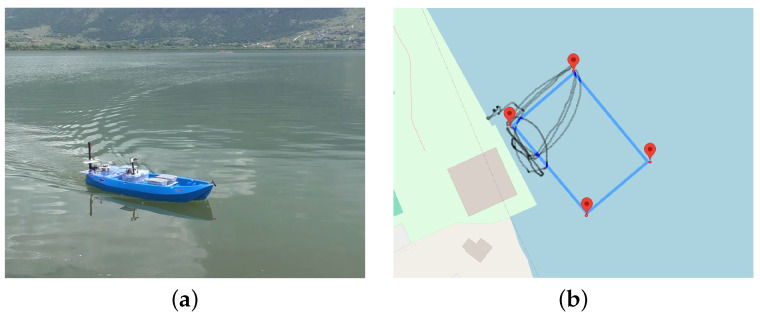
(**a**) Trial at the lake Pamvotis, Ioannina. (**b**) GPS log from the autopilot system test at the lake Pamvotis, Ioannina. On the day of the test there were some pondweed present. Hence, other two waypoints were unreachable. This presented us with the opportunity to test the waypoint skip function.

**Figure 16 sensors-24-03254-f016:**
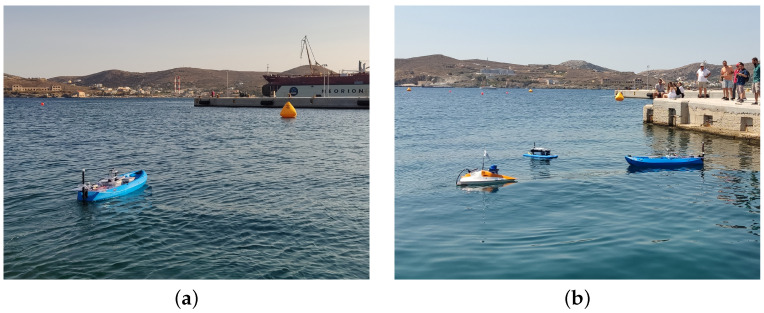
(**a**) Sea trial in Syros, Greece before the race day. (**b**) Ro-boat race competitors on the race day, 12 July 2023.

**Figure 17 sensors-24-03254-f017:**
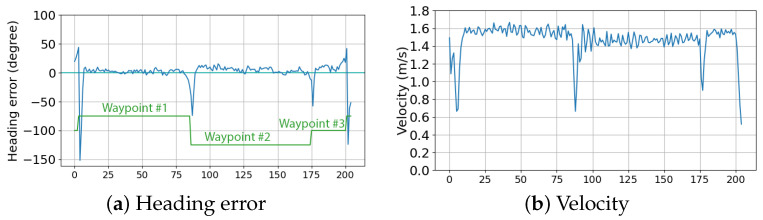
Data collected during the speed race competition: (**a**) heading error of the vessel, and (**b**) velocity recorded during the race in m/s.

**Figure 18 sensors-24-03254-f018:**
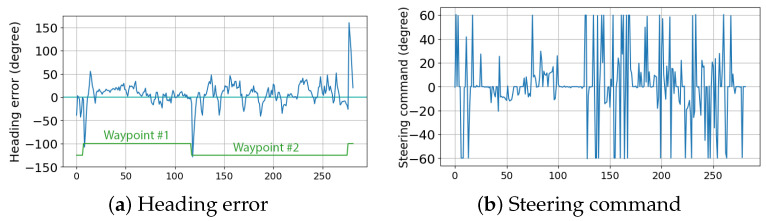
Data collected from the collision avoidance race category: (**a**) heading error of the vessel, and (**b**) steering command received by the collision avoidance algorithm.

**Figure 19 sensors-24-03254-f019:**
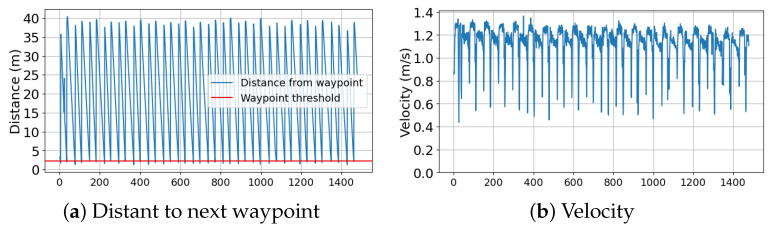
Data collected from the endurance race category: (**a**) distance between the vessel and the next waypoint, and (**b**) velocity of the vessel.

**Figure 20 sensors-24-03254-f020:**
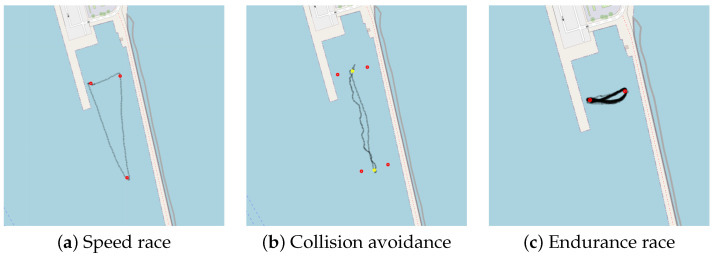
During the 1st Ro-boat race, we collected valuable GPS log data that provide insights into our performance and route throughout the event. The red dots in (**a**–**c**) represent the GPS coordinates given by the race organizer. The yellow dots in (**b**) are the waypoints we entered for the collision avoidance race because the provided GPS coordinates were used as markers for the start/finish and u-turn lines.

**Figure 21 sensors-24-03254-f021:**
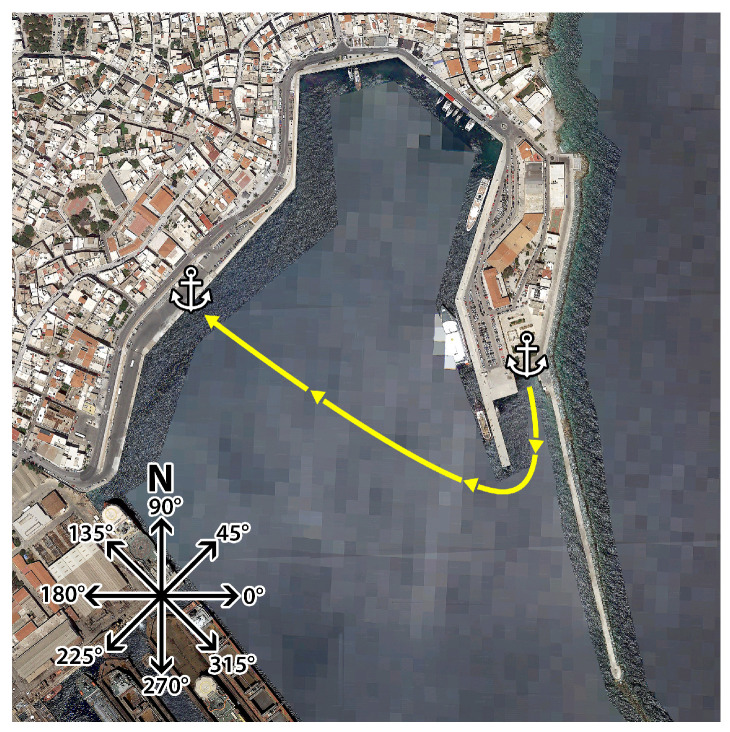
Map of the selected area from Syros, Greece complemented with compass angle marker, starting position and goal used in simulated environment.

**Figure 22 sensors-24-03254-f022:**
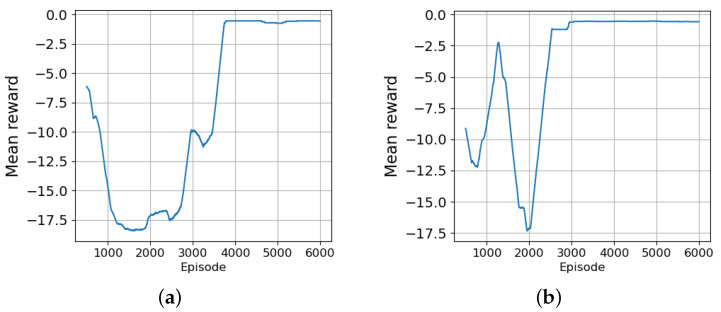
Depiction of the learning curve through the reward in rolling window of 500 episodes. (**a**) Simulated environment with 4 knots wind velocity. (**b**) Simulated environment with 7 knots wind velocity.

**Figure 23 sensors-24-03254-f023:**
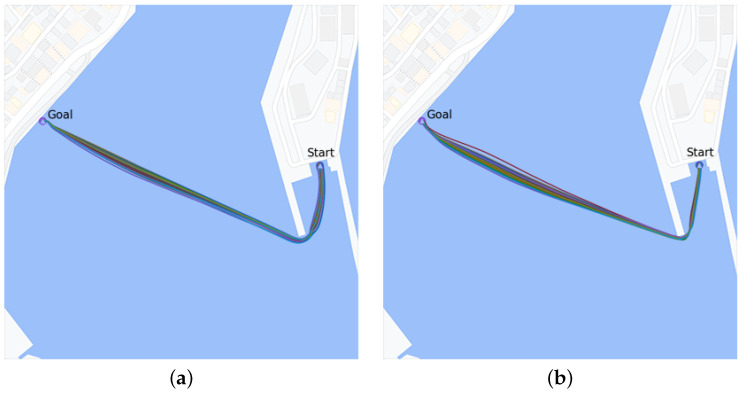
Examples of navigation paths obtained from the learned agent using several environmental disturbances. (**a**) Wind velocity 4 knots, northeast direction. (**b**) Wind velocity 7 knots, northeast direction.

**Table 1 sensors-24-03254-t001:** The specifications of the kayak Lifetime Wave used in this work.

Parameter	Value
Width	61 cm
Height	22 cm
Length	183 cm
Weight	8.2 kg

**Table 2 sensors-24-03254-t002:** Specifications of the propulsion system, consisting of a Haswing W20 electric trolling motor and AGFRC A280BVSW servo.

Parameter	Value
Motor voltage	12 V
Max thrust	20 Lbs
Power-Max	200 W
Model	W20
Prop type	2 blade prop/5.9 inch diameter
Shaft length	60 cm
Pro speed at full power	Max. 1200 rpm underwater
Servo dimensions	65.8 × 30 × 59 mm
Servo weight	284.3 g
Operating voltage	7.4 V–16.8 V
Operating travel	$180^{\circ }\pm 10^{\circ }$
Operating speed	0.08 s/60${\;}^{\circ }$ @16 V
Stall torque	125.0 kg-cm @16 V

**Table 3 sensors-24-03254-t003:** List of the components and the total budget of the vessel.

Item	Manufacturer	Cost (EUR)
Kayak Lifetime Wave	Lifetime Products, Inc. Clearfield, UT, United States	148.8
Trolling Motor Haswing W20	Ya Tai Electric Appliance Co., Ltd., Guangdong, China	178.20
Hobbywing Quicrun WP 880 ESC	Hobbywing Technology Co., Ltd., Shenzhen, China	50
6000 mAh 4S Lipo Battery ×4	Xiamen 3-circles Sports Technology Co.,Ltd., Xiamen, China	163
Plexiglass Sheets	ACRILIX AE, Thessaloniki, Greece	33.12
Nuts, bolts, ball joints, wiring, plugs and misc.	Shenzhen, China	115
Raspberry Pi 4B 8GB	Sony UK Technology Centre, Pencoed, Wales, UK	118.9
Pi Camera	Sony UK Technology Centre, Pencoed, Wales, UK	28.23
Arduino Mega	System Electronica, Scarmagno, Italy	52
Ublox NEO-M8N GPS Module	U-blox, Thalwil, Switzerland	47.03
MPU9250 sensor	SparkFun Electronics, Niwot, CO, United States	14.80
Adafruit SD Card Unit	Adafruit Industries, New York, NY, United States	11.20
Total		960.10

**Table 4 sensors-24-03254-t004:** Evaluation results (mean values and stds) calculated from 1000 executions of the agent’s learned policies in the simulated environment, considering two levels of wind velocity.

Wind Velocity (Knots)	Success Rate (%)	Average Velocity (m/s)	Average Distance (m)
4	100±0	1.71±0.00	493.55±2.31
7	100±0	1.71±0.01	491.82±1.96

## Data Availability

The Arduino sketch and Python script that contain the codes for Algorithms 1 and 2, along with the logs from Ro-boat race portrayed in Figure 17, Figure 18, Figure 19, can be found in the GitHub repository https://github.com/PiyabhumC/Ro-Boat_2023/ (accessed on 5 March 2024).

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
