# Peer review of "Design and Implementation of a Low-Cost Intelligent Unmanned Surface Vehicle"

_sensors, 2024, doi:10.3390/s24103254_

Round 1

Reviewer 1 Report

Comments and Suggestions for Authors

This manuscript provides a detailed account of the design and development of an unmanned surface vehicle (USV) used in a competitive race, emphasizing a low-cost approach utilizing commonly available components. The work presents accessible methods that are likely to be of interest to the robotics research community. The overall contribution of this work appears promising and valuable for dissemination within the field.

However, in reviewing the abstract, a claim is made regarding the utilization of reinforcement learning (RL) strategies as a contribution. However, it is noted that the RL strategies in this work are primarily discussed in the appendix, with the main focus of the manuscript centered on concepts and developments related to RL rather than the specific model and learning strategy of the USV, which I believe has already been presented in their previous work [4]. It is suggested that this claim be reconsidered or appropriately contextualized to maintain focus on the primary contributions of the work.

Furthermore, regarding the collision avoidance algorithm, an important question arises regarding how the authors managed the issue of erroneous detections introduced by MobileNet. Specifically, how were false detections of background objects like isolated mountains or clouds mitigated? Additionally, what impact did such misidentifications have on navigation during experimental trials? Providing clarity on these aspects would enhance the understanding and relevance of the collision avoidance methodology employed.

Author Response

We would like to thank all reviewers for their constructive comments and constructive suggestions. Based on these, the document was further revised, updated and improved. All changes are highlighted (with yellow color) in the revised version of our manuscript. The main changes made are as follows:

  • We improved the introductory section by enriching it with more similar works, and by describing more systematically the contribution of the proposed methodology.
  • We further clarified the description of the obstacle avoidance method to make it easier to understand.
  • We added Section 5 where we describe in more detail how the reinforcement learning agent was constructed and issues about the learning strategy used.
  • We added experiments results (Section 6) from the application of the intelligent agent in simulated environments, and at the same time we improved the presentation of the results obtained in real environments.

Detailed responses to the reviewer’s comments are given below:

This manuscript provides a detailed account of the design and development of an unmanned surface vehicle (USV) used in a competitive race, emphasizing a low-cost approach utilizing commonly available components. The work presents accessible methods that are likely to be of interest to the robotics research community. The overall contribution of this work appears promising and valuable for dissemination within the field.

However, in reviewing the abstract, a claim is made regarding the utilization of reinforcement learning (RL) strategies as a contribution. However, it is noted that the RL strategies in this work are primarily discussed in the appendix, with the main focus of the manuscript centered on concepts and developments related to RL rather than the specific model and learning strategy of the USV, which I believe has already been presented in their previous work [4]. It is suggested that this claim be reconsidered or appropriately contextualized to maintain focus on the primary contributions of the work.

Answer: In the revised version of our manuscript we have modified the format of the text and the layout of the sections to make more clear the contribution of RL to this work. We removed the appendix and included it in the main body of the text as section 5. At the same time, in the experimental results section we have included results about the proposed intelligent agent for navigating the ship in simulated environments in the same sea area where the competition took place. We would also like to clarify that the construction of the agent and the learning process has not been published anywhere else, nor is it related to our recent work [4], which is about imitation learning and movement trajectories provided by experts.

Furthermore, regarding the collision avoidance algorithm, an important question arises regarding how the authors managed the issue of erroneous detections introduced by MobileNet. Specifically, how were false detections of background objects like isolated mountains or clouds mitigated? Additionally, what impact did such misidentifications have on navigation during experimental trials? Providing clarity on these aspects would enhance the understanding and relevance of the collision avoidance methodology employed.

Answer: We have further clarified the usage of the MobileNet in section 4 to emphasize the method we used to detect and identify an object as an obstacle. Regarding the false detections of background objects, we did not train a new model but utilized a pre-trained one. Then we conducted experiments to find out which classes the obstacles presented in the competition would belong to. We only selected the largest obstacle that belongs to the specific classes for the collision avoidance algorithm. Regarding misidentification, the collision detection runs every 2 seconds hence it would constantly look for the next largest object and would identify it eventually. The output of this system is portrayed in Section 6.2, Fig. 18.

Reviewer 2 Report

Comments and Suggestions for Authors

Dear authors!
 I apologize for the delay in publishing my review of your article - I am extremely busy these days.
Your paper looks good and I think there is no essential issues which you should be obligated to fix. As an reviewer, I can add only one remark - I would like to see the sources of your navigation programs for raspberry and sketches for arduino published on github (with links on them in the text). Since you positioning your device as an open source and low cost device, and this, I believe, should also been applied to software, which will allow it to be improved and maintained in the future.
Looking forward to see your paper published soon.
With best wishes,
your reviewer.

Author Response

We would like to thank all reviewers for their constructive comments and constructive suggestions. Based on these, the document was further revised, updated and improved. All changes are highlighted (with yellow color) in the revised version of our manuscript. The main changes made are as follows:

  • We improved the introductory section by enriching it with more similar works, and by describing more systematically the contribution of the proposed methodology.
  • We further clarified the description of the obstacle avoidance method to make it easier to understand.
  • We added Section 5 where we describe in more detail how the reinforcement learning agent was constructed and issues about the learning strategy used.
  • We added experiments results (Section 6) from the application of the intelligent agent in simulated environments, and at the same time we improved the presentation of the results obtained in real environments.

Detailed responses to the reviewer’s comments are given below:

Dear authors!

I apologize for the delay in publishing my review of your article - I am extremely busy these days.
Your paper looks good and I think there is no essential issues which you should be obligated to fix. As an reviewer, I can add only one remark - I would like to see the sources of your navigation programs for raspberry and sketches for arduino published on github (with links on them in the text). Since you positioning your device as an open source and low cost device, and this, I believe, should also been applied to software, which will allow it to be improved and maintained in the future.

Looking forward to see your paper published soon.

Answer: Thank you very much for the help provided to us and for the time you made available despite your heavy schedule. In the revised version we have provided the link to the source code on Github at the end of section 4. The codes are now available at

https://github.com/PiyabhumC/Ro-Boat_2023 .

Reviewer 3 Report

Comments and Suggestions for Authors

In this manuscript, the authors described the design and construction journey of a self-developed unmanned surface vehicle. On this basis, they investigated the corresponding Hardware Architecture, Building Process, Software Architecture, and Experimental Results. This work provides an ideal instrument for validating concepts and algorithms in marine robotics research. The manuscript is well-organized and clearly stated. But there are still some comments that should be taken into consideration, which are presented as follows:

1. The authors may consider increasing the comparative analysis of existing USVs to highlight the advantages of the proposed USVs in terms of cost, performance and applicability.

2. The article does a good job of presenting the technical and application background, but the argument for safety and efficiency improvements might be stronger if it provided some quantitative data or statistical information to support it.

3. For the reinforcement learning strategy used, it is recommended to provide more detailed algorithm description and implementation details to help readers better understand its working principle and effect.

4. The experimental section describes the testing of the system in a static environment, but should include test results in a dynamic environment or real ocean conditions, if possible, to verify the USV's suitability and robustness.

5.The position of Figure A1 is not suitable. It is suggested to adjust the position of Figure A1 in the article.

6.A moderate amount of literature is cited to support the research, but some important citations such as the latest development of relevant technologies can be further updated.

7. Is it possible to extend the proposed results to the Networked Marine Surface Vehicles based on the existed literatures. Please further discuss it.

Comments on the Quality of English Language

Please see the comments attached.

Author Response

We would like to thank all reviewers for their constructive comments and constructive suggestions. Based on these, the document was further revised, updated and improved. All changes are highlighted (with yellow color) in the revised version of our manuscript. The main changes made are as follows:

  • We improved the introductory section by enriching it with more similar works, and by describing more systematically the contribution of the proposed methodology.
  • We further clarified the description of the obstacle avoidance method to make it easier to understand.
  • We added Section 5 where we describe in more detail how the reinforcement learning agent was constructed and issues about the learning strategy used.
  • We added experiments results (Section 6) from the application of the intelligent agent in simulated environments, and at the same time we improved the presentation of the results obtained in real environments.

Detailed responses to the reviewer’s comments are given below:

  1. The authors may consider increasing the comparative analysis of existing USVs to highlight the advantages of the proposed USVs in terms of cost, performance and applicability.

Answer: In the revised version we have included in the introduction section the main advantages of the proposed USV so as to highlight its effectiveness in terms of cost, performance and applicability. In addition, we have added more references related to our work and discussed some recent works concerning the construction of a low-cost USV and enhancing a present USV with low-cost components.

  1. The article does a good job of presenting the technical and application background, but the argument for safety and efficiency improvements might be stronger if it provided some quantitative data or statistical information to support it.

Answer: In section 6.1 of the revised manuscript, we have further emphasized the role and the effectiveness of the safety feature put in place, namely the waypoint skipping feature. We have also added the information about the operation time obtained from the trials.

Regarding statistical information, we have added the log data obtained from the competition along with the analysis of the data (fig. 17-19). The logs consist of velocity, heading error and the distance to the next waypoint to portrait the efficacy of the USV design and effectiveness of the algorithm and its implementation. Regarding the statistical information about improved safety, we have included the recording of collision avoidance command obtained during the race to depict the effectiveness of the collision avoidance system alongside the ability to traverse to the desired goal.

  1. For the reinforcement learning strategy used, it is recommended to provide more detailed algorithm description and implementation details to help readers better understand its working principle and effect.

Answer: In the revised version of our manuscript we have modified the format of the text and the layout of the sections to make more clear the contribution of RL to the work. We have taken out the appendix and included it in the main body of the text as section 5 trying to give more details about the constructive issues and the learning strategy of the RL-based agent. At the same time, in the experimental results section we have included results about the proposed intelligent agent for navigating the ship in simulated environments in the same sea area where the competition took place.

  1. The experimental section describes the testing of the system in a static environment, but should include test results in a dynamic environment or real ocean conditions, if possible, to verify the USV's suitability and robustness.

Answer: We have clarified this issue in the revised version of our manuscript. The proposed USV is tested extensively in real-world environments in Pamvotis lake, and in the port of Syros island. Moreover, we clarified that, because of our design choices (inexpensive, small and light vessel), the proposed USV is suitable for applications in confined environments, like lakes or ports, where the wind and wave disturbances are limited. Possible applications would be the water quality monitoring, object localization and identification on a lake surface, seabed inspection in ports, marine life monitoring. The development of an USV for the open sea or the ocean is out of the scope of this paper. This is mentioned in the introduction section.

5.The position of Figure A1 is not suitable. It is suggested to adjust the position of Figure A1 in the article.

Answer: In the revised version of our manuscript we have included this figure in the experimental results section, along with other relevant figures from results obtained in a simulated environment.

6.A moderate amount of literature is cited to support the research, but some important citations such as the latest development of relevant technologies can be further updated.

Answer: In the revised version we have updated the bibliography by adding some relevant works presenting recent development of the cost for building such vessels. Regarding the relevant technologies, we have also presented and discussed some recent works concerning the construction of a low-cost USV and enhancing a present USV with low-cost components. Also, we have included some recent works on marine platform navigation with reinforcement learning. Totally, we have added nine (9) recent published works in the list of references.

  1. Is it possible to extend the proposed results to the Networked Marine Surface Vehicles based on the existed literatures. Please further discuss it.

Answer: Thank you for this comment. We have added a new paragraph in the end of the Conclusions section where we discuss this issue.

Round 2

Reviewer 3 Report

Comments and Suggestions for Authors

The authors have addressed most of the concerns. As for comment 6, some references on networked marine surface vehicles should be added  to further support the corresponding discussion in Page 19.

Comments on the Quality of English Language

Please see the comments.

Author Response

In the revised edition we have added four (4) references on the topic of Networked Marine Surface Vehicles and to support it as a potential target for future research. These are the (new) references [12], [23], [6], [15], In the order they appear in the last paragraph of our manuscript.